# Non-crossing quantile regression for deep reinforcement learning

**Fan Zhou, Jianing Wang, Xingdong Feng**[*]
School of Statistics and Management
Shanghai University of Finance and Economics
`zhoufan@mail.shufe.edu.cn;jianing.wang@163.sufe.edu.cn;`
`feng.xingdong@mail.shufe.edu.cn`

## Abstract

Distributional reinforcement learning (DRL) estimates the distribution over future returns instead of the mean to more efficiently capture the intrinsic uncertainty of MDPs. However, batch-based DRL algorithms cannot guarantee the non-decreasing property of learned quantile curves especially at the early training stage, leading to abnormal distribution estimates and reduced model interpretability. To address these issues, we introduce a general DRL framework by using non-crossing quantile regression to ensure the monotonicity constraint within each sampled batch, which can be incorporated with some well-known DRL algorithm. We demonstrate the validity of our method from both the theory and model implementation perspectives. Experiments on Atari 2600 Games show that some state-of-art DRL algorithms with the non-crossing modification can significantly outperform their baselines in terms of faster convergence speeds and better testing performance. In particular, our method can effectively recover the distribution information and thus dramatically increase the exploration efficiency when the reward space is extremely sparse.

## 1 Introduction

Different from value-based reinforcement learning algorithms [16, 21, 22] which entirely focus on the expected future return, distributional reinforcement learning (DRL) [12, 20, 24, 17, 1] also accounts for the intrinsic randomness within a Markov Decision Process [5, 4, 19] by modelling the total return as a random variable. Existing DRL algorithms fall into two broad categories, one of which learns quantile values at a set of pre-defined locations such as C51 [1], Rainbow[10], and QR-DQN [5]. Relaxing the constraint on the value range makes QR-DQN achieve a significant improvement over C51 and Rainbow. One recent study, IQN, proposed by [4], shifts the attention from estimating a discrete set of quantiles to the quantile function. IQN has a more flexible architecture than QR-DQN by allowing quantile fractions to be sampled from a uniform distribution. With sufficient network capacity and infinite number of quantiles, IQN can theoretically approximate the full distribution.

A big challenge to distributional RL methods is how to evaluate the validity of the learned quantile distribution. One common problem of fitting quantile regressions at multiple percentiles is the non-monotonicity of the obtained quantile estimates. Much of this issue can be attributed to the fact that the quantile functions are estimated at different quantile levels separately without applying any global constraints to ensure monotonicity. The crossing phenomenon is significantly enhanced by the limited training samples especially at the early stage of the training process, which also increases the difficulty of model interpretation. Without the non-crossing guarantee, the direction of policy searching may be distorted and the selection of optimal actions greatly varies across training epochs. On the other hand,

---

[*]Corresponding Author

the resulted estimation bias of the distribution variance highly reduces the exploration efficiency when using the left truncated variance as the exploration bonus [15]. Sometimes, the upper quantiles instead of the mean (Q-value) are of specific interests when examining some risk-appetite policy. Crossing issue will lead to awkward interpretation due to the abnormal ranking of the quantile points.

Although the crossing issue has been widely examined by the statistics society [13, 9, 14, 8, 6, 3, 2, 7], no optimal solution exists for estimating a general non-linear quantile distribution. In this paper, we introduce a novel approach to obtain non-crossing quantile estimates within the DRL framework. We first give a formal definition of the Wasserstein minimization with the monotonicity restriction and explore its combination with the distributional Bellman operator in practice. Then we describe the detailed implementation procedure of our method by using non-crossing quantile regressions on some state-of-the-art DRL baselines. We examine the performance of the proposed non-crossing method on Atari 2600 games by comparing with the QR-DQN baseline. We repeat the exploration experiment in [15] to more clearly demonstrate the advantage of our method in approximating the real distributions and achieving a more accurate variance estimation.

## 2  Distributional Reinforcement Learning

We consider a Markov Decision Process (MDP) $(\mathcal{S}, \mathcal{A}, \mathbb{R}, p, \gamma)$, with $\mathcal{S}$ and $\mathcal{A}$ being the state and action space. Let $R : \mathcal{S} \times \mathcal{A} \to \mathbb{R}$ be the reward function, and $\gamma \in [0, 1)$ be a discounted factor. $p : \mathcal{S} \times \mathcal{A} \times \mathcal{S} \to [0, 1]$ is the transition probability from $s$ to $s'$ after taking action $a$. $\pi : \mathcal{S} \times \mathcal{A} \to [0, 1]$ is the stochastic policy, which maps state $s$ to a distribution over $\mathcal{A}$. $Z^\pi(s, a)$ denotes the random variable of cumulative rewards the agent gains from $(s, a)$ by following the policy $\pi$, e.g. $Z^\pi(s, a) := \sum_{t=0}^\infty \gamma^t R(s_t, a_t)$ with $s_0 = s$, $a_0 = a$ and $s_{t+1} \sim p(\cdot|s_t, a_t)$, $a_t \sim \pi(\cdot|s_t)$. The expectation of $Z^\pi(s, a)$ is the state-action value

$$Q^\pi(s, a) := \mathbb{E}_{\pi, p}[Z^\pi(s, a)], \tag{1}$$

which is usually approximated by a neural network in most deep reinforcement learning studies. Temporal difference (TD) methods are widely used by value-based RL methods, such as Deep Q-learning[23, 16], to significantly speed up the learning process through the Bellman operator,

$$\mathcal{T}^\pi Q(s, a) := \mathbb{E}[R(s, a)] + \gamma \mathbb{E}_{s' \sim p, \pi}[Q(s', a')]. \tag{2}$$

The target of DQN is to find a globally optimal policy $\pi^*$ to makes $Q^{\pi^*}(s, a) \geq Q^\pi(s, a)$ hold for all $(s, a)$. It is assumed that the potentially optimal policies share the same optimal state-action value function $Q^*$, which is the unique fixed point of the Bellman optimality operator, where

$$Q^*(s, a) = \mathcal{T}Q^*(s, a) := \mathbb{E}[R(s, a)] + \gamma \mathbb{E}_{s' \sim p}\left[\max_{a'} Q^*(s', a')\right]. \tag{3}$$

To learn the optimal $Q^*$, DQN performs a stochastic gradient descent to minimize the loss function

$$\frac{1}{2}\left[r + \gamma \max_{a'} Q_{\omega^-}(s', a') - Q_\omega(s, a)\right]^2, \tag{4}$$

over samples $(s, a, r, s')$. $\omega^-$ is the target network, which is a copy of $\omega$ and is synchronized with $\omega$ periodically. Similar to (2), we can also define the distributional Bellman operator for $Z(s, a)$,

$$\begin{aligned}
\mathcal{T}^\pi Z(s, a) &\overset{D}{:=} R(s, a) + \gamma Z(s', a'), \\
s' &\sim p(\cdot|s, a), a' \sim \pi(\cdot|s'),
\end{aligned} \tag{5}$$

where $Y \overset{D}{=} U$ denotes the equality of probability laws, that is the random variable $Y$ is distributed according to the same law as $U$. DRL looks into the intrinsic randomness of $Z(s, a)$ and repeatedly applies the following distributional Bellman optimality operator [1] in model training

$$\mathcal{T}Z(s, a) \overset{D}{:=} R(s, a) + \gamma Z\left(s', \arg\max_{a' \in \mathcal{A}} \mathbb{E}Z(s', a')\right), s' \sim p(\cdot|s, a). \tag{6}$$

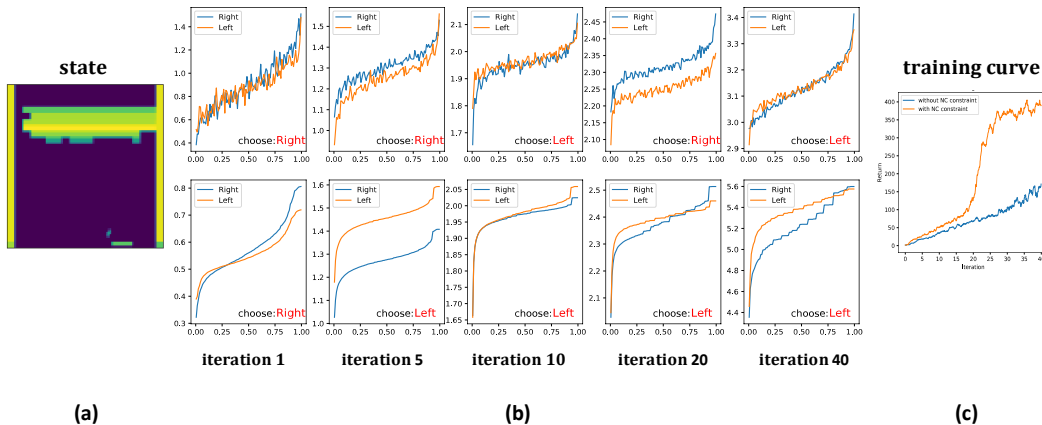

Figure 1: An Atari example to illustrate the crossing issue in DRL training: (a) A representative state $s$ from *Breakout* (b) QR-DQN training with (lower line) or without (upper line) the non-crossing restriction at five different iterations (c) Training curves of two models

## 3 Non-Crossing Quantile Regression for DRL

In this section, we present our method for Distributional RL by using non-crossing quantile regression. We begin by discussing the crossing issue in DRL and the resulting training problems. Then we give the formal definition of the proposed DRL algorithm with the monotonicity restriction, and describe its detailed implementation in practice based on the QR-DQN architecture. Furthermore, we show how the constrained model can more precisely capture the distribution information and increase the efficiency of the exploration strategy proposed by [15] via a real-world Atari example.

### 3.1 Crossing Issue

One important prerequisite to ensure the validity of DRL is that the learned quantile function is non-decreasing, which requires a higher quantile point to have a larger corresponding value. Unfortunately, quantile curves obtained in empirical studies always encounter the crossing issues, leading to the non-monotone quantile estimates. From the theoretical perspective, any $\tau$-th quantile estimate of the return variable $Z$ can converge to its true value as the training size $n$ goes to $\infty$. However, it is impractical for a batch-based DRL algorithm to train the agent using a sufficiently large sample especially in the first few iterations of the training process. On the other hand, since the quantile regression at multiple levels are individually fitted without applying a globally non-crossing constraint, the estimates are unable to quickly approach the true values in case of a large variance. The cumulative deviation makes the sample-based distribution estimate far away from its population-level groundtruth.

Based on the empirical results obtained by training QR-DQN in the Atari environment, *Breakout*, we explain how the crossing issue can lead to the instability of action selections and the accompanying reduced training efficiency. As Figure 1(a) shows, we randomly pick a representative state $s$, in which the small ball is extremely close to the top left of the bat (agent), and the optimal action for the agent to take at this moment is moving to *left*. The lower and upper lines in Figure 1(b) plot the estimated quantile curves for two candidate actions at various training iterations. The only difference is that the lower one takes the non-crossing restriction into consideration when training the QR-DQN model. As Figure 1(b) shows, the standard QR-DQN cannot distinguish from the two ambiguous actions at early iterations due to the severe crossings. A precise estimate of the distribution mean cannot be easily achieved using the abnormal quantile curves. As a comparison, our method effectively addresses this issue, making the quantile function monotonically increasing. With more separate quantile curves of the two actions, the agent can make consistent decisions when perceiving this state in the training process after a few iterations, which to some extent increases the training efficiency. As demonstrated in Figure 1 of the supplements, the crossing issue is more severe with smaller sample size $N$ ($N = 100$) at the early stage, where the advantage of the proposal is more significant. To further show how the ranking of Q-function changes on a variety of states, we randomly pick

4000 states, and compute the probabilities that QRDQN or our method choose the same action with the optimal policy within each of four different training periods. As indicated in Figure 2 of the supplements, our method performs much more stably especially in the early training stage with an overall higher consistency with the optimal policy.

## 3.2 Wasserstein Minimization with Non-Crossing Restriction

We now give the formal definition of the non-crossing quantile regression within the DRL framework. Let $\tilde{\tau} = (\tau_0, \dots, \tau_N)$ be a fixed sequence of $N+1$ non-decreasing quantile fractions, and $\mathcal{Z}_\Theta$ be some pre-defined function space. The goal of DRL is to find an ideal parametric model $\theta : \mathcal{S} \times \mathcal{A} \to \mathbb{R}^N$ to approximate the quantile distribution of $Z$ by using $Z_\theta \in \mathcal{Z}_\Theta$, where

$$Z_\theta(s,a) := \sum_{i=0}^{N-1} (\tau_{i+1} - \tau_i)\,\delta_{\theta_i(s,a)}, \tag{7}$$

is represented by a mix of $N$ supporting quantiles. $\delta_x$ here is a Dirac at $x \in \mathbb{R}$. By definition [18], each $\theta_i(s,a)$ is an estimation of the inverse function of cumulative distribution function (CDF) $F_Z^{-1}(\hat{\tau}_i)$ at the quantile level $\hat{\tau}_i = \frac{\tau_i + \tau_{i+1}}{2}$ with $0 \le i \le N-1$, where

$$F_{Z(s,a)}^{-1}(\tau) := \inf\{z \in \mathbb{R} : \tau \le F_{Z(s,a)}(z)\}. \tag{8}$$

In this case, the theoretically optimal approximation of the return variable $Z$ can be achieved by minimizing a constrained $p$-Wasserstein metric,

$$W_p(Z(s,a), Z_\theta(s,a)) = \left( \sum_{i=0}^{N-1} \int_{\tau_i}^{\tau_{i+1}} \left| F_{Z(s,a)}^{-1}(\omega) - \theta_i(s,a) \right|^p d\omega \right)^{1/p}, \tag{9}$$
$$\text{s.t. } \theta_i(s,a) \le \theta_{i+1}(s,a), \forall\, 0 \le i \le N-2,$$

which effectively measures the difference between the approximated quantile function and the true quantile function $F_{Z(s,a)}^{-1}$ under the non-crossing restriction. When $p$ goes to $\infty$, $W_\infty(Z(s,a), Z_\theta(s,a)) = \sup_{i,\omega \in [\tau_i, \tau_{i+1}]} \left| F_{Z(s,a)}^{-1}(\omega) - \theta_i(s,a) \right|$. Unfortunately, the constraint in (9) is ignored by all the existing DRL algorithms. With the high dimensional representation obtained at each state-action pair $(s,a)$, the crossing becomes even more problematic as the curves have a much larger space $\mathcal{Z}_\Theta$ in which they may cross. Minimizing the unconstrained $p$-Wasserstein metric within an enlarged searching space may result in an infeasible optimal solution given a finite sample. When the sample size goes to infinity, the solution to the constrained optimization in (9) is exactly the solution to the unconstrained one. (9) can be easily solved if $\theta_i$ is either a linear or a simple non-parametric model [9, 2]. For some complex tasks such as Atari learning, it is difficult to find the optimal solution under the original space $\mathcal{Z}_\Theta$ due to the complicated structure of $\theta_i$. To solve this problem, we search for a sub-space $\mathcal{Z}_Q \subset \mathcal{Z}_\Theta$, in which each parametric model satisfies the non-crossing restriction. Let $q : \mathcal{S} \times \mathcal{A} \to \mathbb{R}^N$ such that $Z_q \in \mathcal{Z}_Q$, and $q_i(s,a)$ be the corresponding quantile estimate at $\hat{\tau}_i$. We can re-define (7) as

$$Z_q(s,a) := \sum_{i=0}^{N-1} (\tau_{i+1} - \tau_i)\,\delta_{q_i(s,a)}. \tag{10}$$

In this case, solving (9) has been transferred to finding a projection operator $\Pi_{W_1}$, such that

$$\Pi_{W_1} Z := \underset{Z_q \in \mathcal{Z}_Q}{\arg\min} W_1(Z, Z_q). \tag{11}$$

Given the reduced function space $\mathcal{Z}_Q$, we can obtain some similar results to [5], in which the combination of the projection defined in (11) with Bellman operator is a contraction, i.e.

**Lemma 1.** *Let $\Pi_{W_1}$ be the quantile projection defined in (11), and when applied to value distributions gives the projection for each state-value distribution. For any two value distributions $Z_1, Z_2 \in \mathcal{Z}$ for an MDP with countable state and action spaces,*

$$\bar{d}_\infty \left( \Pi_{W_1} \mathcal{T}^\pi Z_1, \Pi_{W_1} \mathcal{T}^\pi Z_2 \right) \le \gamma \bar{d}_\infty (Z_1, Z_2), \tag{12}$$

*where*

$$\bar{d}_p(Z_1, Z_2) := \sup_{s,a} W_p(Z_1(s,a), Z_2(s,a)), \tag{13}$$

*and $\mathcal{Z}$ be the space of action-value distributions with finite moments:*

$$\mathcal{Z} = \{Z : \mathcal{S} \times \mathcal{A} \to \mathscr{P}(\mathbb{R}) | \; \mathbb{E}\left[|Z(s,a)|^p\right] < \infty, \forall (s,a), p \geq 1\}. \tag{14}$$

Lemma 1 illustrates that the combination operator $\Pi_{W_1}\mathcal{T}^\pi$ is a $\gamma$-contraction under the measure $\bar{d}_\infty$, and the repeated application of this operator converges to a **fixed point** in space $\mathcal{Z}_Q$, denoted as $Z_q^*$. We now introduce the following theorem to give the properties of this fixed point.

**Theorem 1.** *The fixed point $Z_q^*$ is of the form as given in (10) with each quantile $q_i$ satisfying the following equality*

$$q_i(s,a) = R(s,a) + \gamma q_i(s', a'), \; 0 \leq i \leq N-1, \\ s' \sim p(\cdot|s,a), a' \sim \pi(\cdot|s'), \tag{15}$$

*where $\pi$ is a given policy. Let $\Pi_{W_1}Z^\pi = \sum_{i=0}^{N-1}(\tau_{i+1} - \tau_i)\delta_{\bar{q}_i(s,a)}$, with $\bar{q}_i$ being the $\hat{\tau}_i$-th quantile of $Z^\pi$, we can obtain that*

$$Z_q^* \overset{D}{=} \Pi_{W_1}Z^\pi. \tag{16}$$

*When $N \to \infty$, we further have*

$$\bar{d}_\infty(Z^\pi, Z_q^*) \to 0 \; \text{ and } \; Z_q^* \to Z^\pi \text{in distribution.} \tag{17}$$

By Theorem 1, $Z_q^*$ converges to $Z^\pi$ given the policy $\pi$, so the proposed algorithm can be used to choose the optimal policy that maximizes the cumulative reward.

### 3.3 Model Implementation using Non-Crossing Quantile Regression

We can now propose the complete algorithmic approach to ensure monotonicity of the learned quantile curves. We choose QR-DQN as the baseline to describe the implementation details, while the theoretical framework introduced in the previous section can be applied to any quantile based DRL algorithms. The modified model $\mathcal{Z}_q$ is named by NC-QR-DQN, and belongs to the sub-space $\mathcal{Z}_Q$. Two major components of NC-QR-DQN include a *Non-crossing Quantile Logit Network* which generates a sequence of non-decreasing logits based on the state-action representation and a *Scale Factor Network* to recover the logits in $[0,1]$ to the original quantile range. Following

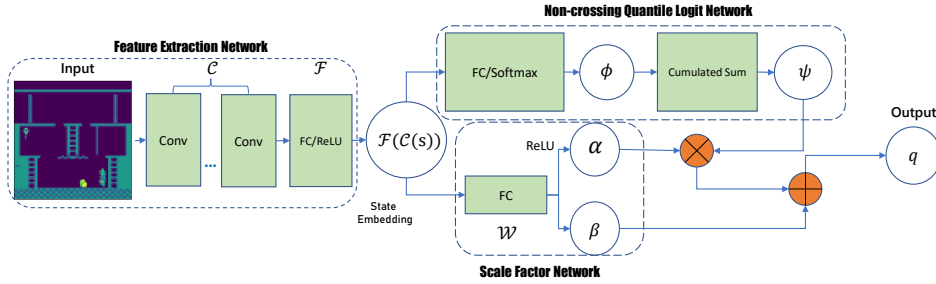

Figure 2: A general picture of the NC-QR-DQN architecture

the main structure of QR-DQN, the *Feature Extraction Network*, which consists of a multi-layer convolution operator $\mathcal{C}$ and a subsequent fully-connected layers $\mathcal{F}$, serves as a feature extractor to capture the embedding $\mathcal{F}(\mathcal{C}(s)) \in R^d$ of state $s$. Then a novel *Non-crossing Quantile Logit Network* $\phi$ maps $\mathcal{F}(\mathcal{C}(s))$ to an $N * |\mathcal{A}|$-dimensional logits by using a fully-connected layer with softmax transformation, i.e. $\phi : \mathbb{R}^d \to [0,1]^{N*|\mathcal{A}|}$. Then we employ a cumulated sum layer to make *Non-crossing Quantile Logit Network* have the final output sorted in a non-decreasing order for each $a$. To be specific, we let $\phi_{i,a}$ be the $i$-th element of the obtained $N$-dimensional logits for action $a$,

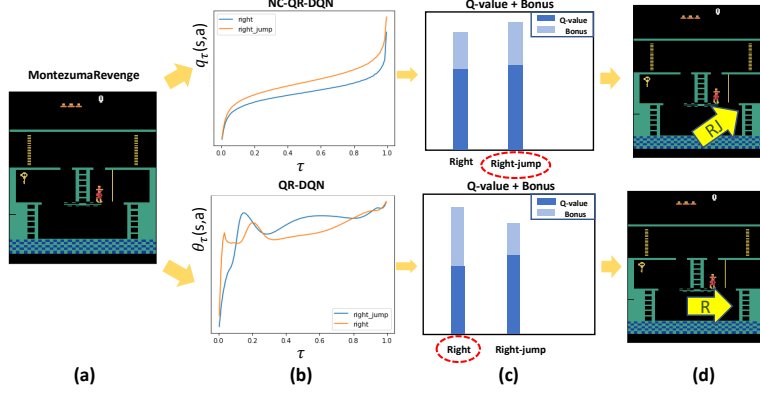

Figure 3: An Atari example to show how the crossing issue may affect the exploration efficiency

such that $\phi_{i,a} = \phi(\mathcal{F}(\mathcal{C}(s)))_{i,a} \in (0,1)$, and $\sum_{i=0}^{N-1} \phi_{i,a} = 1$. Motivated by [25], we calculate the cumulative sum of $\phi_{i,a}$'s from 1 to $i$ each time, and derive a sequence of non-decreasing fractions,

$$\psi_{i,a} := \sum_{j=0}^{i} \phi_{j,a}, \ i = 0, \ldots, N-1; a = 1, \ldots, |\mathcal{A}|. \tag{18}$$

Since all the $N * |\mathcal{A}|$ logits obtained by *Non-crossing Quantile Logit Network* fall into $[0,1]$, we need another transformation to rescale them to a comparable range with the true quantile values. To achieve this target, we introduce the *Scale Factor Network* which calculates two adapted scale factors $\alpha(s,a)$ and $\beta(s,a)$ by applying a fully connected layer $\mathcal{W} : \mathbb{R}^d \to \mathbb{R}^{|\mathcal{A}|*2}$ onto the state embedding $\mathcal{F}(\mathcal{C}(s))$. The flexible architecture of $\mathcal{W}$ allows the range of $\alpha$ and $\beta$ to be varied with the input state and action. In particular, we impose a ReLU function on each $\alpha_a(s)$ to ensure its non-negativity. The final quantile estimates $q_i(s,a)$'s under the non-crossing restriction can be defined as follows,

$$q_i(s,a) := \alpha(s,a) \times \psi_{i,a} + \beta(s,a), i = 0, \ldots, N-1; a = 1, \ldots, |\mathcal{A}|. \tag{19}$$

Since $\alpha(s,a)$ is non-negative and $\{\psi_{i,a}\}$'s are non-decreasing, the non-crossing property of the $N$ $q_i(s,a)$'s is automatically satisfied. Accordingly, we can obtain a modified distributional TD error as

$$\delta_{ij} = r + \gamma q_j\left(s', a^*\right) - q_i(s,a), \quad \forall i, j, \tag{20}$$

where $a^* = \arg\max_{a'} \sum_{j=0}^{N-1} q_j\left(s', a'\right)$. To minimize the Wasserstein metric between the quantile estimates and the target distribution, we train the NC-QR-DQN network by minimizing the Huber quantile regression loss [11],

$$\frac{1}{N} \sum_{i=0}^{N-1} \sum_{j=0}^{N-1} \rho_{\hat{\tau}_i}^{\kappa}(\delta_{ij}), \tag{21}$$

where

$$\begin{aligned} \rho_\tau^\kappa\left(\delta\right) &= \left|\tau - \mathbb{I}\left\{\delta < 0\right\}\right| \frac{\mathcal{L}_\kappa\left(\delta\right)}{\kappa}, \text{ with} \\ \mathcal{L}_\kappa\left(\delta\right) &= \begin{cases} \frac{1}{2}\delta^2, & \text{if } |\delta| \leq \kappa \\ \kappa\left(|\delta| - \frac{1}{2}\kappa\right), & \text{otherwise} \end{cases} \end{aligned} \tag{22}$$

### 3.4 Exploration using truncated variance

We know the target of DRL is to sufficiently model the full distribution of future returns and precisely capture its intrinsic uncertainty. However, some state-of-the-art DRL algorithms, such as QR-DQN and IQN, only use the distribution mean at the future state $s'$ to decide the optimal action $a^*$ when calculating the TD error in (20). To more sufficiently utilize the distribution information, Marvin et al. [15] proposes a novel exploration strategy, Decaying Left Truncated Variance (DLTV), for QR-DQN by using Left Truncated Variance of the estimated quantile distribution as a bonus term to encourage the searching in an unknown space. The key idea behind this design is that the quantile

distribution is usually asymmetric while the upper tail variability is more relevant than the lower tail for instantiating optimism considering the uncertainty. To increase the stability, they use the left truncated measure of the variable, $\sigma_+^2$, based on the median $\tilde{\theta}$ rather than the mean,

$$\sigma_+^2 = \frac{1}{2N} \sum_{i=N/2}^{N-1} \left( \tilde{\theta} - \theta_i \right)^2, \tag{23}$$

where $\theta_i$ is the quantile estimation at quantile level $\hat{\tau}_i$. The exploration bonus is allowed to decay at the following rate to suppress the intrinsic uncertainty:

$$c_t = c\sqrt{\log t/t}, \tag{24}$$

where $t$ is the count of time steps. The optimal action $a^*$ at state $s$ is selected according to

$$a^* = \arg\max_{a'} \left( Q(s, a') + c_t \sqrt{\sigma_+^2} \right). \tag{25}$$

As mentioned in Section 3.1, the crossing issue may highly influence the stability in calculating distribution mean $Q(s, a')$, while the estimation of $\sigma_+^2$ can be even more severely biased. On one hand, the median $\tilde{\theta}$ could be either smaller or larger than its groundtruth due to the crossing. On the other hand, the deviation of the $N/2$ upper-tail quantile estimates from their true values can jointly increase the estimation bias of $\sigma_+^2$. For example, the quantile fractions within a small range of the median but larger than $0.5$ may have a smaller quantile estimate than $\tilde{\theta}$, such that its contribution in $\sigma_+^2$ can be either overestimated or underestimated.

Figure 3(a) plots a randomly sampled state $s$ from $Montezuma Revenge$, with two candidate actions, *right* and *right jump*, selected from 18 feasible actions in total. From the human perspective, the optimal action to take is *right jump* at this moment, which allows the agent to catch the rope and get a higher chance to find the key in the future. Figure 3(b) illustrates the quantile estimates for the two actions obtained from NC-QR-DQN and QR-DQN. Due to the severe crossings in QR-DQN, the intrinsic uncertainty of *right* is overestimated. In this case, $\sigma_+^2$ is too large to be ignored in (25) although the mean value $Q(s, a')$ of *right* is smaller than that of *right jump*. By replacing $\theta_i$ in (23) with $q_i$ in (21), the crossing problem is no longer present and the quantile curves are smoothed as the upper plot of Figure 3(b) shows. The action *right jump* is preferred with both larger mean return $Q$ and bonus $\sigma_+^2$, which accords with the intuition of human beings. In summary, a precise estimation of the truncated variance by including the non-crossing constraint can effectively measure the intrinsic uncertainty and inspire the agent to try actions with high returns in the upper tail.

## 4   Experiments

We test our method on the full Atari-57 benchmark. We select QR-DQN as the baseline, and compare it with NC-QR-DQN which accounts for the non-crossing issue by using the implementation approach described in Section 3.3. We adopt the DLTV exploration strategy proposed in [15] for both algorithms to more effectively assess their performances in modelling the return distributions. We build the QR-DQN baseline using PyTorch, and implement NC-QR-DQN based on the same *Feature Extraction Network* architecture of QR-DQN. We set the number of quantiles $N$ to be 200 and evaluate both algorithms on 200 million training frames. We follow all the parameter settings of [5] and initialize the learning rate to be $5 \times 10^{-5}$ at the training stage. For the exploration set-up, we set the bonus rate $c_t$ in (25) to be $50\sqrt{\log t/t}$ which decays with the training step $t$. For both algorithms, we set $\kappa = 1$ for the Huber quantile loss in (22) due to its smoothness.

The main results are summarized in Table 1, which provides the comparisons between our method with some baseline methods according to their best, human-normalized, scores starting with 30 random no-op actions for each of the 57 Atari games. We can see that NC-QR-DQN significantly outperforms QR-DQN in terms of both the higher mean and median metrics. A detailed raw score table for a single seed across all games, starting with 30 no-op actions is provided in the supplements. To show how the non-crossing fix helps improve the exploration efficiency especially at the early training stage, we compare the testing scores of NC-QR-DQN + $exploration$ and QR-DQN + $exploration$ on all the 57 Atari games within 40M training frames. An improvement score reported

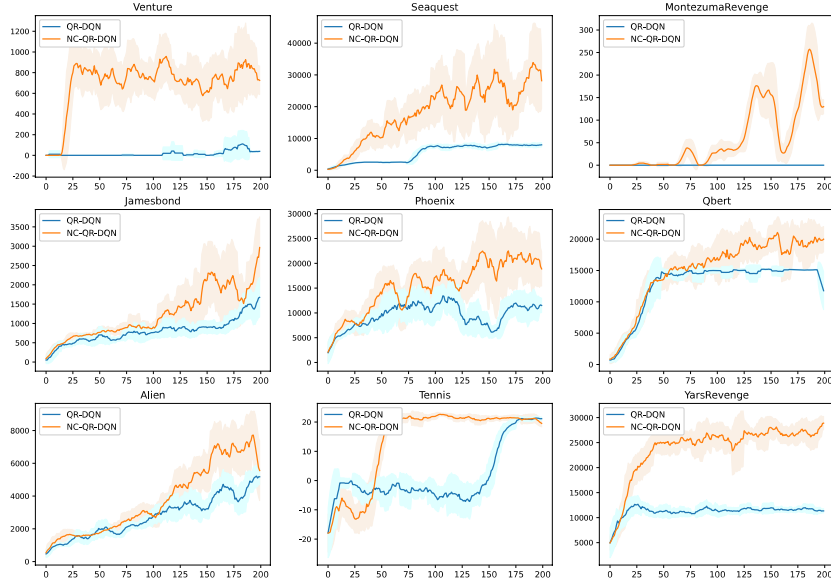

Figure 4: Performance comparison with QR-DQN. Each training curve is averaged by seeds.

in Figure 5(a) is given by the following formula,

$$score = \frac{agent1 - random}{agent2 - random} - 1 \qquad (26)$$

where $agent1, agent2$ and $random$ are the per-game raw scores derived from NC-QR-DQN + $exploration$, QR-DQN + $exploration$ and random agent baseline. As Figure 5(a) shows, NC-QR-DQN + $exploration$ either significantly outperforms its counterpart or achieves a very close result for most of the 57 cases, which verifies our assumption that, NC-QR-DQN can more precisely learn the quantile functions, and highly increase the exploration efficiency by considering the non-crossing restriction. Figure 4 shows the training curves of 9 Atari games averaged by seeds, and we can see that NC-QR-DQN with exploration can learn much faster than QR-DQN with exploration by addressing the crossing issue especially for three hard-explored games presented in the first line.

| Model | Mean | Median | >human |
|---|---|---|---|
| DQN | 228% | 79% | 24 |
| PR.DUEL | 592% | 124% | 39 |
| QR-DQN | 864% | 193% | 41 |
| NC-QR-DQN | **1598%** | **222%** | **42** |

Table 1: Mean and median of scores across 57 Atari 2600 games, measured as percentages of human baseline. Scores are averages over number of seeds.

In practice, NC-QR-DQN is roughly 1% - 5% slower than QR-DQN per training iteration across 57 Atari games, which can be ignored considering its extraordinary outperformance. To be more specific, we compare the computation costs of the two methods in Figure 5. To be fair, we exclude the first five games in Figure 5(b), and we observe that NC-QR-DQN in average is about 30% faster than QR-DQN to achieve the training return at the same level.

## 5 Conclusion

In this work, we introduce a novel space contraction method by making use of global information to ensure the batch-based monotonicity of the learned quantile function by modifying the network architecture of some state-of-the-art DRL algorithms. Although the empirical success it achieves in Atari games especially in the exploration experiments, there are still some open questions to be answered. First, our method is theoretically correct for any DRL algorithms based on quantile

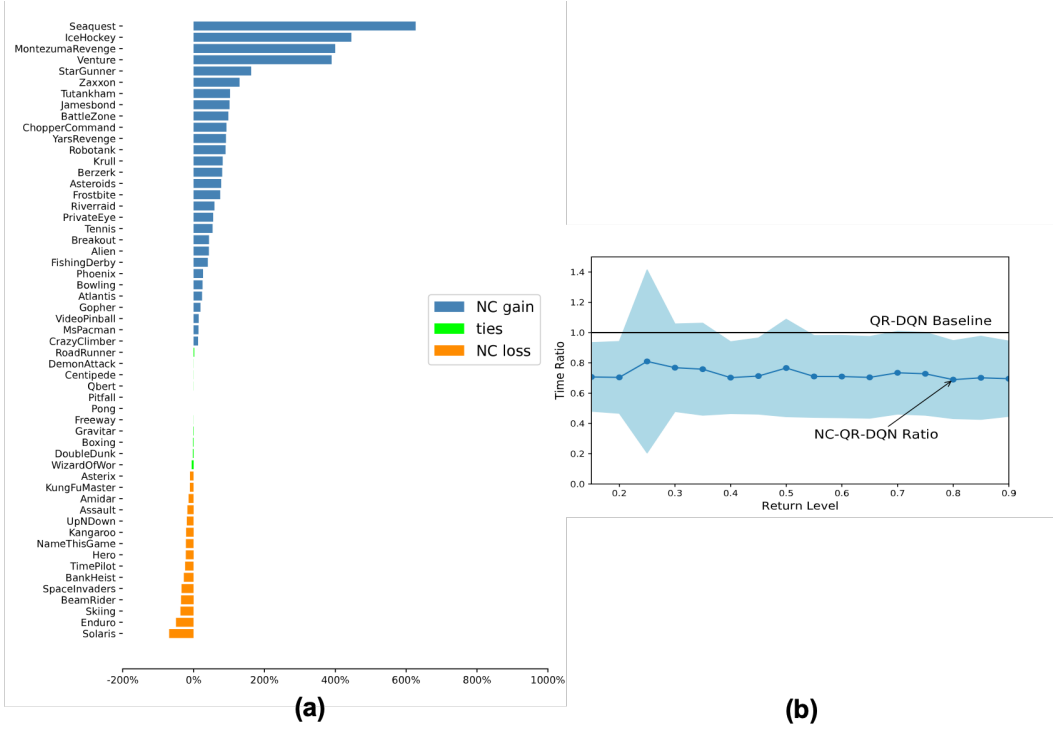

Figure 5: (a) Return improvement of NC-QR-DQN compared to baseline QR-DQN with exploration; (b) Computation cost comparison between NC-QR-DQN and QR-DQN

approximation, while the implementation approach in this paper can not be directly applied to some distribution based methods, such as IQN, since the quantile fractions $\tau$'s are not fixed and re-sampled each time. Second, we indirectly solve the constrained optimization in (9) by changing the model space, while there may exist some other solutions in practice.

## Broader Impact

This work has broad social impact because reinforcement learning is useful in many applied areas including automatic car driving, industrial robotics, and so on. The proposed method on distributional reinforcement learning can more precisely capture the intrinsic uncertainty of MDPs by ensuring the non-crossing of quantile estimates, which helps AI to better understand some complicated real-world decision making problems. Our method highly increases the exploration efficiency of DRL algorithms, and can be widely used in some difficult tasks that have extremely large state-reward spaces. On the other hand, allowing the agent to explore more uncertainty of the environment may change the way robots think and lead to some negative outcomes in real-life.

## Acknowledgments and Disclosure of Funding

This research was supported by National Natural Science Foundation of China (12001356, 11971292, 11690012), Shanghai Sailing Program (20YF1412300), Fundamental Research Funds for the Central Universities, and Program for Innovative Research Team of SUFE.

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
