[Supplementary Material]

# Supplements of "Non-crossing quantile regression in deep reinforcement learning"

**Fan Zhou, Jianing Wang, Xingdong Feng**
School of Statistics and Management
Shanghai University of Finance and Economics
zhoufan@mail.shufe.edu.cn;jianing.wang@163.sufe.edu.cn;
feng.xingdong@mail.shufe.edu.cn

## 1  Proof of Lemma 1

We first introduce the following Lemma , which is used to complete the proof of Lemma 1.

**Lemma.** *Consider an MDP with countable state and action spaces. Let $Z_1, Z_2$ be value distributions such that each state-action distribution of $Z_1(s,a)$ or $Z_2(s,a)$ is a single Dirac. Consider the particular case where rewards are identically 0, and let $\tau \in [0,1]$. Denote by $\Pi_\tau$ the projection operator that maps a probability distribution onto a Dirac delta located as its $\tau$-th quantile. Then*

$$\bar{d}_\infty \left( \Pi_\tau \mathcal{T}^\pi Z_1, \Pi_\tau \mathcal{T}^\pi Z_2 \right) \leq \gamma \bar{d}_\infty \left( Z_1, Z_2 \right), \tag{1}$$

*Proof.* The proof is similar to the argument of that of Lemma 3 of [1]. Let $Z_1(s,a) = \delta_{q_{(s,a)}}$ and $Z_2(s,a) = \delta_{\psi_{(s,a)}}$ for each state-action pair $(s,a) \in \mathcal{S} \times \mathcal{A}$, for some functions $\psi, q : \mathcal{S} \times \mathcal{A} \to \mathbb{R}$. Let $(s', a')$ be a state-action pair, and let $((s_i, a_i))_{i \in I}$ be all the state-action pairs that are accessible from $(s', a')$ in a single transition, with $I$ an indexing set. To simplify notations, we write $q_i$ for $q(s_i, a_i)$ and $\psi_i$ for $\psi(s_i, a_i)$. Furthermore, let the probability of transiting from $(s', a')$ to $(s_i, a_i)$ be $p_i$, for all $i \in I$.

Then we have

$$\begin{aligned} (\mathcal{T}^\pi Z_1)(s', a') &= \gamma \sum_{i \in I} p_i \delta_{q_i}, \\ (\mathcal{T}^\pi Z_2)(s', a') &= \gamma \sum_{i \in I} p_i \delta_{\psi_i}. \end{aligned} \tag{2}$$

Now consider the $\tau$-th quantile of each of these distributions, for $\tau \in [0,1]$ arbitrary. Let $u \in I$ be the index such that $q_u$ is the $\tau$-th quantile of $\sum_{i \in I} p_i \delta_{q_i}$, and let $v \in I$ be the index such that $\psi_v$ is the $\tau$-th quantile of $\sum_{i \in I} p_i \delta_{\psi_i}$. Thus, we obtain that

$$\bar{d}_\infty \left( \Pi_\tau \mathcal{T}^\pi Z_1, \Pi_\tau \mathcal{T}^\pi Z_2 \right) = \gamma |q_u - \psi_v|. \tag{3}$$

We now show that the inequality

$$|q_u - \psi_v| \leq |q_i - \psi_i|, \quad \forall i \in I, \tag{4}$$

holds, by which it follows that

$$\bar{d}_\infty \left( \Pi_\tau \mathcal{T}^\pi Z_1(s', a'), \Pi_\tau \mathcal{T}^\pi Z_2(s', a') \right) \leq \gamma \bar{d}_\infty \left( Z_1, Z_2 \right), \tag{5}$$

and the result of Lemma 1 then follows by taking maxima over state-action pairs $(s', a')$.

To obtain the inequality (4), without loss of generality we take $q_u \leq \psi_v$. We now introduce the following partitions of the indexing set $I$. Let

$$
\begin{aligned}
I_{\leq q_u} &= \{i \in I | q_i \leq q_u\}, \\
I_{>q_u} &= \{i \in I | q_i > q_u\}, \\
I_{<\psi_v} &= \{i \in I | \psi_i < \psi_v\}, \\
I_{\geq \psi_v} &= \{i \in I | \psi_i \geq \psi_v\},
\end{aligned}
\tag{6}
$$

and we then have the following disjoint unions:

$$
\begin{aligned}
I &= I_{\leq q_u} \cup I_{>q_u} \\
I &= I_{<\psi_v} \cup I_{\geq \psi_v}.
\end{aligned}
\tag{7}
$$

If the inequality (4) does not hold, then we must have $I_{\leq q_u} \cap I_{\geq \psi_v} = \emptyset$. It then follows that $I_{\leq q_u} \subseteq I_{<\psi_v}$. Thus, since $q_u$ is the $\tau$-th quantile of $\sum_{i \in I} p_i \delta_{q_i}$, we obtain that

$$
\sum_{i \in I_{\leq q_u}} p_i \geq \tau,
\tag{8}
$$

and so consequently

$$
\sum_{i \in I_{<\psi_v}} p_i \geq \tau,
\tag{9}
$$

which implies that the $\tau$-th quantile of $\sum_{i \in I} p_i \delta_{\psi_i}$ is less than $\psi_v$, and leads to a contraction. Therefore, the inequality (4) holds, which completes the proof. $\square$

Now we give the proof of Lemma 1.

**Lemma 1.** *Let $\Pi_{W_1}$ be the quantile projection defined as above, and when applied to value distributions gives the projection for each state-value distribution. For any two value distributions $Z_1, Z_2 \in \mathcal{Z}$ for an MDP with countable state and action spaces,*

$$
\bar{d}_\infty \left( \Pi_{W_1} \mathcal{T}^\pi Z_1, \Pi_{W_1} \mathcal{T}^\pi Z_2 \right) \leq \gamma \bar{d}_\infty \left( Z_1, Z_2 \right),
\tag{10}
$$

*where*

$$
\bar{d}_p(Z_1, Z_2) := \sup_{s,a} W_p(Z_1(s,a), Z_2(s,a)),
\tag{11}
$$

*and $\mathcal{Z}$ be the space of action-value distributions with finite moments:*

$$
\mathcal{Z} = \{Z : \mathcal{S} \times \mathcal{A} \to \mathscr{P}(\mathbb{R}) | \, \mathbb{E}\left[|Z(s,a)|^p\right] < \infty, \forall(s,a), p \geq 1\}.
\tag{12}
$$

*Proof.* The proof is similar to the argument of that of Proposition 2 of [1]. We assume that instantaneous rewards given a state-action pair are deterministic, and the general case is a straight-forward generalization with the regular probability argument. Furthermore, since Wasserstein distances are invariant under translation of the support of distributions, it is sufficient to consider the case where $r(s, a) \equiv 0$ for all $(s, a) \in \mathcal{S} \times \mathcal{A}$. The proof then proceeds by first considering the case where every value distribution consists only of single Diracs based on the result of Lemma 1.

We write $Z_1(s, a) = \sum_{k=0}^{N-1} \frac{1}{N} \delta_{q_k(s,a)}$ and $Z_2(s, a) = \sum_{k=0}^{N-1} \frac{1}{N} \delta_{\psi_k(s,a)}$, where the functions $q, \psi : \mathcal{S} \times \mathcal{A} \to \mathbb{R}^N$ are shape-constrained for ensuring non-crossing quantiles. Let $(s, a)$ be a state-action pair, and let $((s_i, a_i))_{i \in I}$ be all the state-action pairs that are accessible from $(s', a')$ in a single transition, where $I$ is a indexing set. Write $p_i$ for the probability of transitioning from $(s', a')$ to $(s_i, a_i)$, for each $i \in I$. We now construct a new MDP and new value distributions for this MDP in which all distributions are given by single Diracs, with a view to applying Lemma 1. The new MDP is of the following form. We take the stat-action pair $(s', a')$, and define new states, actions, transitions and a policy $\widetilde{\pi}$, so that the state-action pairs accessible from $(s', a')$ in this new MDP are given by $((\widetilde{s}_i^j, \widetilde{a}_i^j)_{i \in I})_{j=0}^{N-1}$, and the probability of reaching the state-action pair $(\widetilde{s}_i^j, \widetilde{a}_i^j)$ is $p_i/N$. Furthermore, we define new value distributions $\widetilde{Z}_1, \widetilde{Z}_2$ as follows. For each $i \in I$ and $j = 0, \ldots, N-1$, we consider

$$
\begin{aligned}
\widetilde{Z}_1 \left( \widetilde{s}_i^j, \widetilde{a}_i^j \right) &= \delta_{q_j(s_i,a_i)} \\
\widetilde{Z}_2 \left( \widetilde{s}_i^j, \widetilde{a}_i^j \right) &= \delta_{\psi_j(s_i,a_i)}.
\end{aligned}
\tag{13}
$$

Since the $\bar{d}_\infty$ distance between the 1-Wasserstein projections of two real-valued distributions is the max over the difference of a certain set of quantiles, we may appeal to Lemma 1 to obtain the following result:

$$
\begin{aligned}
& \bar{d}_\infty \left( \Pi_{W_1} \left( \mathcal{T}^{\widetilde{\pi}} \widetilde{Z}_1 \right)(s',a'), \Pi_{W_1} \left( \mathcal{T}^{\widetilde{\pi}} \widetilde{Z}_2 \right)(s',a') \right) \\
& \leq \gamma \sup_{\substack{i \in I \\ j=0,\ldots,N-1}} |q_j(s_i,a_i) - \psi_j(s_i,a_i)| \\
& = \gamma \sup_{i \in I} \bar{d}_\infty \left( Z_1(s_i,a_i), Z_2(s_i,a_i) \right).
\end{aligned}
\tag{14}
$$

Now note that by construction, $(\mathcal{T}^{\widetilde{\pi}} \widetilde{Z}_1)(s',a')$ has the same distribution as $(\mathcal{T}^{\pi} Z)(s',a')$, and thus we have

$$
\begin{aligned}
& \bar{d}_\infty \left( \Pi_{W_1}(\mathcal{T}^{\widetilde{\pi}} \widetilde{Z}_1)(s',a'), \Pi_{W_1}(\mathcal{T}^{\widetilde{\pi}} \widetilde{Z}_2)(s',a') \right) \\
& = \bar{d}_\infty \left( \Pi_{W_1}(\mathcal{T}^{\pi} Z_1)(s',a'), \Pi_{W_1}(\mathcal{T}^{\pi} Z_2)(s',a') \right).
\end{aligned}
\tag{15}
$$

Therefore, substituting this into (14), we obtain

$$
\begin{aligned}
& \bar{d}_\infty \left( \Pi_{W_1}(\mathcal{T}^{\pi} Z_1)(s',a'), \Pi_{W_1}(\mathcal{T}^{\pi} Z_2)(s',a') \right) \\
& \leq \gamma \sup_{i \in I} \bar{d}_\infty \left( Z_1(s_i,a_i), Z_2(s_i,a_i) \right).
\end{aligned}
\tag{16}
$$

Taking suprema over the initial state $(s',a')$ then yields the result.

$\square$

## 2  Proof of Theorem 1

**Theorem 1.** *The fixed point $Z_q^*$ is of the form as $Z_q^*(s,a) := \sum_{i=0}^{N-1} (\tau_{i+1} - \tau_i) \delta_{q_i(s,a)}$ with each quantile $q_i$ satisfying the following equality*

$$
\begin{aligned}
q_i(s,a) &= R(s,a) + \gamma q_i(s',a'), \ \ 0 \leq i \leq N-1, \\
s' &\sim p(\cdot|s,a), a' \sim \pi(\cdot|s'),
\end{aligned}
\tag{17}
$$

*where $\pi$ is a given policy. Let $\Pi_{W_1} Z^{\pi} = \sum_{i=0}^{N-1} (\tau_{i+1} - \tau_i) \delta_{\bar{q}_i(s,a)}$, with $\bar{q}_i$ being the $\hat{\tau}_i$-th quantile of $Z^{\pi}$,*

*we can obtain that*

$$
Z_q^* \overset{D}{=} \Pi_{W_1} Z^{\pi}.
\tag{18}
$$

*When $N \to \infty$, we further have*

$$
\bar{d}_\infty(Z^{\pi}, Z_q^*) \to 0 \ \ and \ \ Z_q^* \to Z^{\pi} \ in \ distribution.
\tag{19}
$$

*Proof.* Assume that the instantaneous rewards are deterministic given a stat-action pair and the total return $Z^{\pi}$ has a continuous CDF $F_{Z^{\pi}}(z)$, which can also be generalized to the random case. For $\epsilon > 0$, let $\tau_0 = \epsilon, \tau_N = 1 - \epsilon$ and $\tau_0, \ldots, \tau_N$ are equidistant fractions, and $\hat{\tau}_i = \frac{\tau_i + \tau_{i+1}}{2}$. We firstly verify that $Z_q^*$ is the fixed point of $\Pi_{W_1} \mathcal{T}^{\pi}$. In other words, we need to show that $\Pi_{W_1} \mathcal{T}^{\pi} Z_q^* = Z_q^*$. For any state-action pair $(s,a)$, $(s',a')$ is accessible from $(s,a)$ in a single transition, by the definition of $q_i$, which is the $\hat{\tau}_i$-th quantile value, we have

$$
\begin{aligned}
& P \left( Z_q^*(s',a') \leq q_i(s',a') \right) \\
& = P \left( R(s,a) + \gamma Z_q^*(s',a') \leq R(s,a) + \gamma q_i(s',a') \right) \\
& = \hat{\tau}_i.
\end{aligned}
\tag{20}
$$

By (17), we obtain that $P \left( R(s,a) + \gamma Z_q^*(s',a') \leq q_i(s,a) \right) = \hat{\tau}_i$. Note that $P \left( Z_q^*(s,a) \leq q_i(s,a) \right) = \hat{\tau}_i$. We then get that $Z_q^*(s,a) \overset{D}{=} R(s,a) + \gamma Z_q^*(s',a')$ on each quantile fraction. Thus $\mathcal{T}^{\pi} Z_q^* = Z_q^*$ holds. On the other hand, it is clear that $Z_q^*$ is an element of $\mathcal{Z}_Q$, then the fixed point result follows.

Furthermore, due to the fact that $\bar{q}_i$ is the $\hat{\tau}_i$-th quantile of $Z^\pi$, we have

$$P\left(Z^\pi(s,a) \leq \bar{q}_i(s,a)\right) = \hat{\tau}_i. \tag{21}$$

Recall the definition of $Z^\pi$, we then have

$$Z^\pi(s,a) = R(s,a) + \gamma Z^\pi(s',a'), \text{for all } s' \sim P(\cdot|s,a), a' \sim \pi(\cdot|s'). \tag{22}$$

By (21) and (22), we obtain that

$$
\begin{aligned}
&P\left(Z^\pi(s,a) \leq \bar{q}_i(s,a)\right) \\
=&P\left(R(s,a) + \gamma Z^\pi(s',a') \leq \bar{q}_i(s,a)\right) \\
=&P\left(Z^\pi(s',a') \leq \left(\bar{q}_i(s,a) - R(s,a)\right)/\gamma\right) \\
=&\hat{\tau}_i, \text{ for all } s' \sim P(\cdot|s,a), a' \sim \pi(\cdot|s').
\end{aligned}
$$

Therefore,

$$\bar{q}_i(s,a) = R(s,a) + \gamma\bar{q}_i(s',a'), \text{ for all } s' \sim P(\cdot|s,a), a' \sim \pi(\cdot|s'). \tag{23}$$

Due to the uniqueness of fixed point, we have $Z_q^* \overset{D}{=} \Pi_{W_1} Z^\pi$.

At last, it is straight to show $\bar{d}_\infty(Z^\pi, \Pi_{W_1} Z^\pi) \to 0$ as $N \to \infty$. In fact, the monotonicity of $F_{Z^\pi}^{-1}(\tau)$ implies that

$$\bar{d}_\infty(Z^\pi, \Pi_{W_1} Z^\pi) = \sup_i \left(\max\left(\left|F_{Z^\pi}^{-1}(\tau_i) - F_{Z^\pi}^{-1}(\hat{\tau}_i)\right|, \left|F_{Z^\pi}^{-1}(\tau_{i+1}) - F_{Z^\pi}^{-1}(\hat{\tau}_i)\right|\right)\right). \tag{24}$$

Since the quantile function $F_{Z^\pi}^{-1}(\tau)$ is uniformly continuous on $[\epsilon, 1-\epsilon]$ because the distribution function $F_{Z^\pi}(z)$ is assumed to be continuous, therefore, let $N \to \infty$, we have $|\hat{\tau}_i - \tau_i| \to 0$ and $|\hat{\tau}_i - \tau_{i+1}| \to 0$, then $\max(\left|F_{Z^\pi}^{-1}(\tau_i) - F_{Z^\pi}^{-1}(\hat{\tau}_i)\right|, \left|F_{Z^\pi}^{-1}(\tau_{i+1}) - F_{Z^\pi}^{-1}(\hat{\tau}_i)\right|) \to 0$, for each $i = 0, \ldots N - 1$, the result follows.

For $\forall z \in \left(F_{Z^\pi}^{-1}(\tau_0), F_{Z^\pi}^{-1}(\tau_N)\right)$, we could find the index $i$ such that $F_{Z^\pi}^{-1}(\hat{\tau}_i) \leq z \leq F_{Z^\pi}^{-1}(\hat{\tau}_{i+1})$. Then $|F_{Z_q^*}(z) - F_{Z^\pi}(z)| \leq |\hat{\tau}_{i+1} - \hat{\tau}_i| \to 0$ as $N \to \infty$. Thus $Z_q^*$ converges to $Z^\pi$ in distribution. $\square$

# 3   Figures 1 and 2

Figure 1: Training comparison between NC-QR-DQN and QR-DQN with N = 100 and 200 on Breakout at different training stages

Figure 2: Boxplot of the probabilities that QRDQN or NC-QRDQN chooses the same action with the optimal policy for 4000 randomly selected states within each of four different training period

# 4 Raw Score table across all Atari games

| GAMES | RANDOM | HUMAN | DQN | PRIOR. DUEL. | QR-DQN | NC-QR-DQN |
|---|---|---|---|---|---|---|
| Alien | 227.8 | 7,127.7 | 1,620.0 | 3,941.0 | 4,871 | **10,277.4** |
| Amidar | 5.8 | 1,719.5 | 978.0 | 2,296.8 | 1,641 | **2,031.5** |
| Assault | 222.4 | 742.0 | 4,280.4 | 11,477.0 | **22,012** | 21,766.5 |
| Asterix | 210.0 | 8,503.3 | 4,359.0 | 375,080.0 | **261,025** | 148,681.1 |
| Asteroids | 719.1 | 47,388.7 | 1,364.5 | 1,192.7 | **4,226** | 2,824.8 |
| Atlantis | 12,850.0 | 29,028.1 | 279,987.0 | 395,762.0 | 971,850 | **1,015,973.1** |
| BankHeist | 14.2 | 753.1 | 455.0 | 1,503.1 | 1,249 | **1,357.5** |
| BattleZone | 2,360.0 | 37,187.5 | 29,900.0 | 35,520.0 | 39,268 | **55,675.6** |
| BeamRider | 363.9 | 16,926.5 | 8,627.5 | 30,276.5 | **34,821** | 22,619.4 |
| Berzerk | 123.7 | 2,630.4 | 585.6 | 3,409.0 | 3,117 | **170,386** |
| Bowling | 23.1 | 160.7 | 50.4 | 46.7 | 77.2 | **95.9** |
| Boxing | 0.1 | 12.1 | 88.0 | 98.9 | **99.9** | **99.9** |
| Breakout | 1.7 | 30.5 | 385.5 | 366.0 | 742 | **749** |
| Centipede | 2,090.9 | 12,017.0 | 4,657.7 | 7,687.5 | **12,447** | 10,206.9 |
| ChopperCommand | 811.0 | 7,387.8 | 6,126.0 | 13,185.0 | **14,667** | 10,458.3 |
| CrazyClimber | 10,780.5 | 35,829.4 | 110,763.0 | 162,224.0 | 161,196 | **178,325.0** |
| DemonAttack | 152.1 | 1,971.0 | 12,149.4 | 72,878.6 | 121,551 | **122,737.0** |
| DoubleDunk | -18.6 | -16.4 | -6.6 | -12.5 | 21.9 | **22** |
| Enduro | 0.0 | 860.5 | 729.0 | 2,306.4 | **2,355** | 2,342.6 |
| FishingDerby | -91.7 | -38.7 | -4.9 | 41.3 | **39.0** | 37.4 |
| Freeway | 0.0 | 29.6 | 30.8 | 33.0 | 34.0 | **34.0** |
| Frostbite | 65.2 | 4,334.7 | 797.4 | 7,413.0 | 4,384 | **6,463.5** |
| Gopher | 257.6 | 2,412.5 | 8,777.4 | 104,368.2 | **113,585** | 82,954.2 |
| Gravitar | 173.0 | 3,351.4 | 473.0 | 238.0 | 995 | **1,007.5** |
| Hero | 1,027.0 | 30,826.4 | 20,437.8 | 21,036.5 | 21,395 | **29,397** |
| IceHockey | -11.2 | 0.9 | -1.9 | -0.4 | -1.7 | **-0.8** |
| Jamesbond | 29.0 | 302.8 | 768.5 | 812.0 | 4,703 | **8,552** |
| Kangaroo | 52.0 | 3,035.0 | 7,259.0 | 1,792.0 | 15,356 | **16,987.5** |
| Krull | 1,598.0 | 2,665.5 | 8,422.3 | 10,374.4 | **11,447** | 9,493.8 |
| KungFuMaster | 258.5 | 22,736.3 | 26,059.0 | 48,375.0 | **76,642** | 53,644 |
| MontezumaRevenge | 0.0 | 4,753.3 | 0.0 | 0.0 | 0.0 | **330.8** |
| MsPacman | 307.3 | 6,951.6 | 3,085.6 | 3,327.3 | 5,821 | **6,149** |
| NameThisGame | 2,292.3 | 8,049.0 | 8,207.8 | 15,572.5 | **21,890** | 18,657.1 |
| Phoenix | 761.4 | 7,242.6 | 8,485.2 | 70,324.3 | 16,585 | **32,797** |
| Pitfall | -229.4 | 6,463.7 | -286.1 | 0.0 | **0.0** | **0.0** |
| Pong | -20.7 | 14.6 | 19.5 | 20.9 | **21.0** | **21.0** |
| PrivateEye | 24.9 | 69,571.3 | 146.7 | 206.0 | **350** | 200 |
| Qbert | 163.9 | 13,455.0 | 13,117.3 | 18,760.3 | **572,510** | 25,317.9 |
| RiverRaid | 1,338.5 | 17,118.0 | 7,377.6 | 20,607.6 | 17,571 | **19,545.4** |
| RoadRunner | 11.5 | 7,845.0 | 39,544.0 | 62,151.0 | 64,262 | **69,738** |
| Robotank | 2.2 | 11.9 | 63.9 | 27.5 | 59.4 | **71.6** |
| Seaquest | 68.4 | 42,054.7 | 5,860.6 | 931.6 | 8,268 | **62,300** |
| Skiing | -17,098.1 | -4,336.9 | -13,062.3 | -19,949.9 | -9,324 | **-9,034.1** |
| Solaris | 1,236.3 | 12,326.7 | 3,482.8 | 133.4 | **6,740** | 2,140 |
| SpaceInvaders | 148.0 | 1,668.7 | 1,692.3 | 15,311.5 | **20,972** | 12,166.3 |
| StarGunner | 664.0 | 10,250.0 | 54,282.0 | 125,117.0 | 77,495 | **146,337.5** |
| Tennis | -23.8 | -8.3 | 12.2 | 0.0 | 23.6 | **23.8** |
| TimePilot | 3,568.0 | 5,229.2 | 4,870.0 | 7,553.0 | **10,345** | 8,145.6 |
| Tutankham | 11.4 | 167.6 | 68.1 | 245.9 | 297 | **358** |
| UpNDown | 533.4 | 11,693.2 | 9,989.9 | 33,879.1 | **71,260** | 34,886.1 |
| Venture | 0.0 | 1,187.5 | 163.0 | 48.0 | 43.9 | **1,481** |
| VideoPinball | 16,256.9 | 17,667.9 | 196,760.4 | 479,197.0 | **705,662** | 561,229.6 |
| WizardOfWor | 563.5 | 4,756.5 | 2,704.0 | 12,352.0 | 25,061 | **26,359.2** |
| YarsRevenge | 3,092.9 | 54,576.9 | 18,098.9 | 69,618.1 | 26,447 | **31,260.1** |
| Zaxxon | 32.5 | 9,173.3 | 5,363.0 | 13,886.0 | **13,112** | 11,954.3 |

# References

[1] Will Dabney, Mark Rowland, Marc G Bellemare, and Rémi Munos. Distributional reinforcement learning with quantile regression. In *Thirty-Second AAAI Conference on Artificial Intelligence*, 2018.