[Reviews · NeurIPS 2020]

Review 1

Summary and Contributions: This paper contributes: 1. Identification of a major issue in quantile-based distributional RL algorithms, i.e. crossing quantiles. 2. A network architecture that resolves the issue. 3. Evaluation of the proposed architecture. 4. Integrating the architecture with DLTV.

Strengths: - Soundness: Good. The idea of having non-crossing quantiles is very straightforward yet not investigated in existing works. I believe the non-crossing property would greatly improve all quantile-based distributional RL algorithms. - Significance and novelty: Medium. The idea is based on existing algorithms. Evaluation results seem to bring significant improvements on QR-DQN, but QR-DQN itself is a relatively early work in the family of distributional RL algorithms.

Weaknesses: - Baseline algorithm: While all quantile-based distributional RL algorithms suffer from the crossing quantile issue, QR-DQN is the least affected one since the quantiles are uniformly fixed. IQN[1], which uses randomly sampled quantiles, and FQF[2], which optimizes over chosen quantiles for better distribution approximation, are both expected to suffer much more from crossing quantiles than QR-DQN. While it may be non-trivial to adapt NC architecture to IQN since the quantiles are randommly sampled, it shouldn't be hard to adapt to FQF. Besides, IQN and FQF both have achieved much higher scores than QR-DQN, hence I believe implementing NC architecture on IQN and FQF would greatly strenghthen empirical validations. - Evaluated environments: Most RL works that use ALE for evaluation include results for all 57 games. Can authors explain why only 49 out of 57 games are used for evaluation? - Number of quantiles: I believe that N=100 quantiles is a reasonable choice. But since the original QR-DQN paper used N=200, a comparison under the same setting should be included as well. I suspect that lower number of quantiles would suffer more from the crossing issue, if the authors can commit to providing detailed evaluation results over different number of quantiles I would definitely provide a stronger recommendation to this paper. - Hard exploration games: The authors claim that NC method "highly increase the exploration efficiency by considering the non-crossing restriction", however this claim is only validated on 21 games where "in which the QR-DQN baseline with DLTV does not perform well". I would love to see complete results on full 57 games, especially hard exploration games[3] such as Motezuma's Revenge, Pitfall and Private Eye. References: [1] Dabney, Will, et al. "Implicit Quantile Networks for Distributional Reinforcement Learning." International Conference on Machine Learning. 2018. [2] Yang, Derek, et al. "Fully parameterized quantile function for distributional reinforcement learning." Advances in Neural Information Processing Systems. 2019. [3] Aytar, Yusuf, et al. "Playing hard exploration games by watching youtube." Advances in Neural Information Processing Systems. 2018.

Correctness: Overall the claims and method are correct. Some questions I have in reading the paper include: - Need of \alpha: I didn't understand the need of first scaling logits to [0,1] then multiply it by a non-negative scalar \alpha. How is it different from directly using ReLU transformation after \phi? - QR-DQN implementation: How exactly is the QR-DQN baseline implemented? Are QR-DQN raw scores comparable to the original paper?

Clarity: Overall the paper is well written and easy to read. The main concern I have is on the score table. Throughout the paper and the supplementary material, no raw scores are provided and it is hard to evaluate authors contribution. I hope that the authors could provide a more detailed results including raw scores for all evaluated games, mean and median human normalized scores as well as number of super-human performance agents. Figure 6 is hard to understand at first glance.

Relation to Prior Work: The differences between this work and prior works are clearly discussed.

Reproducibility: Yes

Additional Feedback: --Post author feedback-- I thank the authors for their feedbacks. I will update my score to 6 since the authors commit to provide more detailed experiments and an IQN-version of NC method shows significant improvements. The scores for raw IQN seems off compared with what reported in the original paper, I hope that the authors would clarify this results in their final version.


Review 2

Summary and Contributions: The paper studies the quantile crossing issue in distributional reinforcement learning (DRL). Specifically, the crossing issue refers to the fact that the existing DRL algorithms cannot guarantee the monotonicity of the learned quantile curves (especially at the early training stage), which leads to abnormal distributional estimates and reduces model interpretability. To handle this issue, the authors introduce non-crossing quantile regression to ensure that the monotonicity constraints are satisfied. They demonstrate the validity of the proposed solution through theoretical analysis and empirical evaluation. The experiments show that the proposed method outperforms the baseline, QR-DQN. The main contributions are: (i) This paper identifies the crossing issue that appears in quantile-based DRL, and (ii) resolves this by using a revised architecture for QR-DQN with a cumulated softmax function.

Strengths: 1. Formally address the crossing issue in distributional RL with a clear explanation and illustrative examples 2. Propose an architectural solution to enforce monotonicity of quantiles

Weaknesses: 1. The experimental results of QR-DQN seem questionable: Compared to the results in the original QR-DQN paper (e.g. the results of Enduro, Kangaroo, KungFuMaster, and Hero in Figure 6 of (Dabney et al., AAAI 2018)), the performance of QR-DQN in this paper (Figure 4) looks much worse. Similar observations can be made from the IQN paper (i.e. Dabney et al., ICML 2018). Based on this, the correctness of the experimental evaluation is questionable. 2. The proposed solution is not clearly justified: While the crossing issue does exist in QR-DQN (as shown in Figure 1), from the engineering perspective, it seems much natural to simply sort the atoms before calculating the quantile Huber loss. The paper does not provide enough explanation about why the proposed solution is preferred. 3. Missing reference: The idea of using cumulated softmax is very similar to the Fully Parameterized Quantile Function (FQF) proposed by (Yang et al., NeurIPS 2019). However, this critical reference is missing.

Correctness: The description of the crossing issue is correct. However, as mentioned above, the experimental results of QR-DQN seem questionable.

Clarity: Overall it is clearly written and well-organized.

Relation to Prior Work: One important reference (i.e. FQF) is missing.

Reproducibility: Yes

Additional Feedback: Theorem 1 seems to be a direct result of Proposition 2 in (Dabney et al., AAAI 2018) since the quantile distribution $\mathcal{Z}_\theta$ and the space of quantile distributions $\mathcal{Z}_Q$ in (Dabney et al., AAAI 2018) already implicitly satisfy the non-crossing conditions. If this is the case, then the current presentation of Theorem 1 is misleading as it appears like a new theoretical contribution (instead, Theorem 1 shall be restated as a lemma for the proof of Proposition 1). ===== Post-rebuttal ===== The authors’ response does ease some of my concerns on the design of NC-based methods, but my major concern still remains: the performance of the benchmark QR-DQN looks somewhat questionable as it does not match that of the original paper (also mentioned by R1). However, the authors did not fully address this issue in the rebuttal. Also, similar things can be observed in the new experimental results of IQN provided in the rebuttal letter (e.g., in Breakout and Q*bert, the obtained scores are much worse than those in the original IQN paper). To quantify the severity of the crossing quantile issue, it is crucial to have a fair comparison between the benchmark methods and the NC counterparts.


Review 3

Summary and Contributions: Distributional RL (DRL) uses a set of quantiles to approximate the return distribution, however, the monotonicity of quantiles is not imposed in prior algorithms such as QR-DQN. This paper proposes non-crossing (NC) quantile regression to explicitly impose such constraints, leading to improved exploration with sparse rewards as well as performance gains on Atari 2600 games.

Strengths: > A well motivated and novel solution for fixing the non-crossing inconsistency of the quantile distribution learned by QR-DQN (a popular DRL algorithm). > Good empirical evaluation with observed performance gains especially the experiments related to exploration which use the spread of the distribution rather than only its mean. > Potential to create better distributional algorithms in the future which better approximate distributional RL algorithms.

Weaknesses: > Although the paper is well-motivated with good empirical results, some statements in the introduction and abstract are not validated sufficiently and only weak evidence is provided [see comments section for more details] > [Minor] As stated by the authors, NC-QR is not applicable to more advanced distributional algorithms such as IQN which approximate the continuous quantile function, however the issue of non-monotonicity is also applicable to IQN.

Correctness: Yes.

Clarity: Yes.

Relation to Prior Work: Yes.

Reproducibility: Yes

Additional Feedback: I'd suggest the authors to tone down claims and/or present more empirical evidence for the following statements: > Line 5: "reduced model interpretability" -- Please expand upon how the non-crossing fix improves model interpretability -- the non-crossing criteria improves the distribution approximation, however, what notion of "interpretability" is being used here: a smoothed version of quantile curve for QR-DQN still looks monotonic and doesn't seem to affect interpretability much. > Line 34: "The selection of optimal action greatly varies across training epochs": Since only the expectation of the distribution is used for selecting actions, it is unclear if the non-monotonicity of quantiles would affect the distribution mean much -- Figure 1 only shows one particular state and is not convincing enough -- a plot showing how the ranking of Q-function changes on a variety of states and comparing it with the true ranking (possibly based on the optimal policy) would be more convincing. Questions (and potential improvements): > Are the results based on deterministic Atari? Since learning the distributions might be trickier in stochastic Atari (via sticky actions), it would be great to see results on the stochastic setting too. This would help bolster the claims about the effectiveness of NC QR-DQN over QR-DQN. > Reporting results on standardized setting used on Atari (200M frames of training) would make the paper more impactful and I highly recommend the authors to do so for the updated version and also report a median-normalized performance, so that future approaches can easily compare their results with this paper. Minor Grammar suggestions: -> Line 110: Unfortunately --> However -> Line 230: "NC-QR-DQN in average" --> "NC-QR-DQN, on average," - > Line 174: "more sufficiently" --> "better" > Presentation suggestions: The text size in figures was quite hard to read, please consider using larger text size!


Review 4

Summary and Contributions: This paper presents a variant of quantile regression DQN (QR-DQN) with an additional constraint that enforces monotonicity of the quantiles, a property which is not present in the original QR-DQN algorithm. The authors accompany their new algorithm with theoretical results and compelling empirical evidence. I have read the authors rebuttal, but my score remains unchanged.

Strengths: Well-motivated, new algorithm based on solid theoretical principles which are explained well throughout. The theoretical results seem sound, and the empirical evidence provided is convincing.

Weaknesses: There are two main limitations for me. The first is that no code (nor even pseudocode) was provided with the submission, which is not good for reproducibility. I hope the authors plan on making their code open-source if the paper is accepted. The second is that the atari experiments were not run for the standard 200 million training frames, although I can appreciate that this requires large amounts of compute if running multiple algorithms on multiple games and with multiple seeds.

Correctness: The theoretical claims and empirical evaluations seem correct.

Clarity: The paper is very well written. Some comments below: - In line 21, I don't think you mean to include QR-DQN in the same list as C51 and Rainbow? - In equation (2), the subscript of the second expectation should probably be `a~\pi`. - Line 88, "the small ball is extremely close to the top left of the bat", the ball is moving down I imagine? Worth specifying. - In equation (9) it may be worth giving the constraints line it's own equation number, as you refer to them further in the discussion. - In lines 115-116: "When the sample size goes to infinity, ... is exactly the solution to the unconstrained one." Is this a known result? If so, please add a reference. If not, worth an extra discussion. - The statement in line 122 also doesn't seem like a clear result. - In line 131, do we require Z_Q to be something like a Banach space, to ensure the proper convergence of the operator to the fixed point? - In line 44 of the appendix, "We now construct a new MDP...", a figure to illustrate the new constructed MDP would help the reader. - Nit: On line 191, is the state s really randomly sampled? :) - Line 46 in appendix: "We take the stat*e*-action pair" - Line 66 in appendix: "and deterministic given a stat*e*-action pair"

Relation to Prior Work: The paper is clearly positioned as an extension/improvement over QR-DQN, and the differences are properly discussed.

Reproducibility: No

Additional Feedback:

[Author Response · NeurIPS 2020]

Thank you for all the valuable feedback. One common comment is that some results may not be comparable to the
original paper. Since we are not researchers from the industry and do not have strong computing power, we set the
number of quantiles to be 100 and evaluate all methods using 40 million training frames for the experiment. We
will definitely try to add the **complete comparison (200M, 57 games)** and more evaluation results **(stochastic Atari,**
**different quantile numbers etc)** into the final version of the paper. We are also working on a modified version of the
NC architecture which can be extended to **IQN**, and its comparison with the baseline IQN (using Google 'Dopamine')
are provided in Figure (a) (last two with exploration). Our response to other specific comments are provided as follows.
**Reviewer #1**:
Q1. *Can authors explain why only 49 out of 57 games are used for evaluation?.*
We choose the 49 Atari game (initially proposed by Mnih et al., 2015) since our main contrast DLTV is evaluated using
the same environment. But we will definitely include all the 57 games in the final version.
Q2. *If the authors can commit to providing detailed evaluation results over different number of quantiles.*
We will provide detailed evaluations in the final version of the paper. Figure (b) compares N = 100, 200 on Breakout.
The crossing issue is more severe with smaller N at early stage, and the improvement of our approach is more significant.
Q3. *Hard exploration games: I would love to see complete results on full 57 games.*
Sorry for the misunderstanding. Figure 5(a) in the paper is actually the comparison between our method (with
exploration) and DLTV on all the 49 games. The improvement in Motezuma's Revenge, and Private Eye are 6545%
(60M) and 25% (40M) respectively. We will improve the presentation of this part in the final version.
**Reviewer #2**:

19
Q1. *It seems much natural to simply sort the atoms before calculating the quantile Huber loss.*
As Figure (c) shows, our approach performs better than 'sort' with a higher optimal return. The key reason is that,
by sorting, the same dimension of the network output may be paired to a different quantile location '$\tau$' each time
without employing the non-decreasing constraint, which highly decreases the training stability and efficiency. With the
monotonicity restriction, we can make better use of global information in training.
Q2. *The idea of using cumulated softmax is very similar to the Fully Parameterized Quantile Function (FQF).*
Sorry for ignoring this important reference. We will cite it in the final draft of the paper.
**Reviewer #3**:
Q1. *Please expand upon how the non-crossing fix improves model interpretability.*
Sometimes, the upper quantiles instead of the mean (Q-value) are of specific interests when examining some risk-
appetite policy. Crossing issue will lead to awkward interpretation due to the abnormal ranking of the quantile points.
Also, the non-crossing fix can bring a more precise estimation of distribution variance when doing exploration.
Q2. *A plot showing how the ranking of Q-function changes on a variety of states would be more convincing.*
We randomly pick 4000 states, and compute the probabilities that QRDQN or NC-QRDQN chooses the same action
with the optimal policy within each of the four training period. As the boxplots in Figure (d) shows, our method
performs much more stable especially in the early stage with an overall higher consistency with the optimal policy.
**Reviewer #4**:
Q1. *"When the sample size goes to infinity, ... " and the statement in line 122 don't seem like clear results.*
When $N$ is fixed and sample size goes to infinity, $\theta_i(s, a)$ converges to the real quantile function $F_Z^{-1}$ at level $\hat{\tau}_i$ as
defined in line 105. Thus, the monotonicity of quantile function guarantees the monotonic constraint in equation (9).
The projection operator $\Pi_{W_1}$ is to find a $Z_q$ in non-crossing space $\mathcal{Z}_Q$ to minimize its $W_1$ distance from $Z$, which is
equivalent to finding a model parameterized by $\theta_i$'s under the monotonicity constraint.
Q2. *Do we require $\mathcal{Z}_Q$ to be a Banach space, to ensure the proper convergence of the operator to the fixed point.*
Yes, we need $\mathcal{Z}_Q$ to be a complete space to ensure the proper convergence.

[Meta-Review · NeurIPS 2020]

The strong rebuttal with additional results on NC-IQN swayed multiple initially hesitant reviewers to argue for acceptance, and I concur. The one unresolved concern is about reproducing the baseline results more accurately: I assume this is a matter of codebase/implementation details that does not detract from fair head-to-head comparisons. However, it's worth discussing any discrepancies in more depth in the final version.